# Exploring Stakeholders’ Perceptions of Electronic Personal Health Records for Mobile Populations Living in Disadvantaged Circumstances: A Multi-Country Feasibility Study in Denmark, Ghana, Kenya, and The Netherlands

**DOI:** 10.3390/ijerph22091363

**Published:** 2025-08-29

**Authors:** Paulien Tensen, Maria Bach Nikolajsen, Simeon Kintu Paul, Princess Ruhama Acheampong, Francisca Gaifém, Frederick Murunga Wekesah, Ulrik Bak Kirk, Ellis Owusu-Dabo, Per Kallestrup, Erik Beune, Charles Agyemang, Steven van de Vijver

**Affiliations:** 1Department of Public and Occupational Health, Amsterdam Public Health, Amsterdam UMC, University of Amsterdam, 1105 AZ Amsterdam, The Netherlands; e.j.beune@amsterdamumc.nl (E.B.); c.o.agyemang@amsterdamumc.nl (C.A.); 2Amsterdam Health & Technology Institute, 1105 BP Amsterdam, The Netherlands; svijver@gmail.com; 3Research Unit for General Practice, 8000 Aarhus, Denmark; mbn@ph.au.dk (M.B.N.); ubk@ph.au.dk (U.B.K.); per.kallestrup@ph.au.dk (P.K.); 4African Population and Health Research Center, Nairobi P.O. Box 10787-00100, Kenya; simeonkintu@gmail.com (S.K.P.); fwekesah@aphrc.org (F.M.W.); 5Department of Health Promotion and Disability Studies, School of Public Health, College of Health Sciences, Kwame Nkrumah University of Science and Technology, Kumasi 00233, Ghana; princess.acheampong@knust.edu.gh; 6Research Unit for Global Health, Aarhus University, 8000 Aarhus, Denmark; fgaifem@gmail.com; 7Julius Global Health, Julius Center for Health Sciences and Primary Care, University Medical Center Utrecht, Utrecht University, 3584 CS Utrecht, The Netherlands; 8Department of Public Health, Aarhus University, 8000 Aarhus, Denmark; 9Department of Global Health, School of Public Health, College of Health Sciences, Kwame Nkrumah University of Science and Technology, Kumasi 00233, Ghana; eowusu-dabo.chs@knust.edu.gh; 10Division of Endocrinology, Diabetes and Metabolism, Department of Medicine, Johns Hopkins University School of Medicine, Baltimore, MD 21205, USA; 11Department of General Practice, OLVG Hospital, 1091 HA Amsterdam, The Netherlands

**Keywords:** electronic personal health record, migration, mobile populations, medical data exchange, health equity, continuity of care, health access, digital health, individual interviews

## Abstract

(1) Background: Mobile populations living in disadvantaged circumstances often face disrupted continuity of care due to incomplete or inaccessible health records. This feasibility study explored the perceived usefulness of Electronic Personal Health Records (EPHRs) in enhancing access to and continuity of care for mobile populations across Denmark, Ghana, Kenya, and The Netherlands. (2) Methods: A qualitative study using ninety semi-structured interviews, with multi-level stakeholders ranging from policymakers to mobile individuals, recruited through purposive and convenience sampling. Interview guides and analysis were informed by the Technology Acceptance Model (TAM), and analysis by the Unified Theory of Acceptance and Use of Technology (UTAUT). (3) Results: Stakeholders highlighted the value of improved medical data sharing and ownership and considered EPHRs promising for enhancing care continuity and efficiency. Key concerns included limited digital and health literacy, and data security and privacy, underscoring the need for education and safeguards against inappropriate data sharing. Due to differences in digital readiness and privacy guidelines, a one-size-fits-all EPHR is unlikely to succeed. (4) Conclusions: EPHRs are considered valuable tools to enhance care continuity and increase patient ownership, but they face technical, structural, and social challenges, including data security and varying levels of digital (health) literacy. Successful implementation requires context-sensitive, co-created solutions supported by strong policy frameworks.

## 1. Introduction

Globally, mobile populations face considerable challenges in accessing healthcare and maintaining continuity of care.

The term ‘mobile populations’ follows the definition by the International Organization for Migration (IOM), referring to individuals or groups who, whether forcibly or voluntarily, change their location of residence [1]. This includes movements that take place both locally, nationally, or across borders for both short- and long-term durations. This article focuses specifically on populations that face more significant disadvantages, including displaced individuals, refugees, asylum seekers, and undocumented migrants (UDMs), henceforth referred to as ‘mobile populations’. Among the challenges faced by mobile populations, a key issue is the limited access to health information due to missing, incomplete, or unavailable health records [2,3,4]. This is often because many healthcare systems worldwide use health records systems that are neither transferable nor interconnected. While limited access to health records affects the general population, it presents a significant barrier to the continuity of care for mobile populations. This is not only because mobile individuals frequently encounter new healthcare providers, but also due to the compounding effect of other disadvantages they face, including legal, administrative, economic, and systemic barriers [5,6,7].

Without access to previous medical history, mobile populations often experience delayed access to healthcare, delayed treatment, discontinuity of care, and disproportionate burden of diseases, such as higher rates of preventable diseases, untreated infectious diseases, and unmonitored chronic diseases [8,9,10].

No well-functioning operational solution has yet been implemented to ensure global data interoperability across healthcare facilities and international borders. While initiatives such as the EU Health Data Space, proposed by the European Commission, are currently being developed to enhance medical data exchange between EU countries, they focus exclusively on individuals registered within Europe, excluding mobile populations in irregular legal circumstances. Additionally, this initiative does not provide patients with control over their health data.

Electronic Personal Health Records (EPHRs) are a potential solution to improving continuity and promoting access to care for mobile populations in general, and in particular those experiencing disadvantages [11,12,13], aligned with the WHO’s objective of ensuring that refugees and migrants benefit from universal health coverage [14]. As defined by the Markle Foundation, Electronic Personal Health Records are “electronic systems that allow individuals to access, manage, and share their health information in a confidential and secure environment that allows users to coordinate their lifelong health data and share relevant portions with those who need it” [15]. Typically, EPHRs include medical history, treatment plans, medications, allergies, test results, immunization records, and other relevant personal information.

Although research in this field remains limited, existing literature suggests that Electronic Personal Health Records may improve health data exchange along migration routes [11], as the patient often represents the only constant in the care continuum [12]. Moreover, an EPHR could enable patients and healthcare providers (HCPs) to access health data anytime and anywhere, thus enhancing continuity of care [12]. Furthermore, EPHRs can empower patients to take a more active role in health management and decision-making [16].

Previous research in The Netherlands indicates that healthcare providers generally hold a positive attitude towards the development and implementation of an EPHR for mobile populations, particularly for undocumented migrants [13]. However, when technology-supported service models are introduced in real-world settings, challenges such as non-adoption, early abandonment, and failure to scale, spread, and sustain these interventions are common [17]. As described in the NASSS framework [18], these challenges are often driven by multiple and complex factors, which require the use of a multi-level, multi-stakeholder approach when assessing the introduction of an EPHR within different contexts [19]. This study intended to explore the feasibility of introducing Electronic Personal Health Records in a multi-country context and focused specifically on two European countries (Denmark and The Netherlands) and two African countries (Kenya and Ghana). The study draws on two widely recognized theoretical frameworks on technology acceptance: the Technology Acceptance Model (TAM) [20] and the Unified Theory of Acceptance and Use of Technology (UTAUT) [21]. While both models focus on individual-level acceptance of technology, they emphasize key constructs such as perceived usefulness (TAM) or performance expectancy (UTAUT), and perceived ease of use (TAM) or effort expectancy (UTAUT), as well as social influence and facilitating conditions (UTAUT). Our study integrates these core concepts to explore multi-level opportunities and challenges related to the perceived usefulness of Electronic Personal Health Records (EPHRs) among mobile populations across diverse settings.

Common to the four settings is the fact that the feasibility of implementing EPHRs, along with their acceptability and adoption by healthcare systems, providers, and patients, has not yet been thoroughly studied—particularly concerning mobile populations who often experience fragmented care [12]. Therefore, this qualitative feasibility study aimed to investigate multi-level stakeholders’ perceptions on the perceived usefulness of EPHRs in improving access to and continuity of care for mobile populations across the four countries.

## 2. Materials and Methods

### 2.1. Study Design

An exploratory qualitative approach consisting of semi-structured interviews with multi-level stakeholders in Denmark, Ghana, Kenya, and The Netherlands was used in this study. The COREQ checklist was used as a guide to report this study [22] (see Appendix A).

### 2.2. Study Setting

Denmark and The Netherlands have highly digitalized healthcare systems. In Denmark, there is relatively good cross-institutional interoperability between different care facilities, although challenges remain, and further improvements are needed [23,24]. In The Netherlands, many healthcare facilities have different electronic health records (EHRs), leading to fragmentation in the health system and limited medical data sharing. Moreover, strict privacy regulations hamper data sharing between facilities [25]. Both African countries are making strides in the digitalization of their health systems, including rolling out EHRs. However, the implementation of EHRs is still at an early stage and is largely limited to urban areas and privately owned health facilities.

While all four countries formally recognize the right of individuals to access their health records, mobile populations often encounter difficulties in obtaining this access. In Denmark, when officially registered, people can digitally access large parts of their health records. However, there is currently no digital solution for individuals without a personal identification number, such as UDMs and asylum seekers. In The Netherlands, individuals officially registered can access part of their health record through a Personal Health Environment (Dutch abbreviation: PGO), though uptake remains limited [26]. Currently, a small-scale pilot of a PGO for people on the move called HealthEmove is being implemented in Amsterdam, The Netherlands [12]. Although Kenya and Ghana are making progress in improving access to and digitizing health records, significant challenges such as limited health systems interoperability, limited digital infrastructure, non-standardized data protocols, and inadequate user awareness remain [27,28,29].

### 2.3. Study Participants and Recruitment

Based on the characteristics of their interaction with EPHRs and the intention to examine the real-world implementation of an EPHR through a multi-level approach [19], potential participants were classified into three main groups before recruitment: macro, meso, and micro level stakeholders. The macro level stakeholders consisted of individuals working at governmental bodies or non-governmental organizations (NGOs) operating at the national level in the field of migrant care. As most of these individuals had decision-making roles, they could be involved in deciding if and how EPHRs could be implemented. Participants grouped in the meso level were individuals working at NGOs operating within a local context, or healthcare organizations providing care to the mobile populations within a single setting. The micro level consisted of individual HCPs or people from mobile communities who would actively use and interact with an EPHR. The mobile populations of interest varied across countries, reflecting the differences in the mobile groups present in the countries. However, all mobile populations shared the common characteristic of frequently changing geographical locations and often living in disadvantaged circumstances. In Denmark and The Netherlands, the main focus was on UDMs and asylum seekers and (Eastern-) EU labor migrants in The Netherlands. The mobile populations in Kenya included refugees, asylum seekers, pastoralist communities, labor migrants, and UDMs. In Ghana, mobile populations were UDMs, refugees, and international workers.

Participants were purposively sampled based on their roles in decision-making for the macro level, or their affiliation with specific (international) NGO’s active across the four countries, such as IOM, the United Nations High Commissioner for Refugees (UNHCR), and the Red Cross. For the meso and micro levels, participants were purposively sampled based on their role at local NGOs or their role as a healthcare provider for mobile groups or mobile populations from the network of researchers. Convenience sampling was used through snowball techniques to extend the poll of potential participants as they were recruited. All participants in The Netherlands and Denmark were invited via email, while in Ghana and Kenya, participants were invited either via email, a phone call, or a physical encounter. A total of 96 participants were invited to participate in the study, and 90 participants were eventually involved in the study. The reasons for not participating in the interviews were due to no response to (follow-up) emails.

### 2.4. Data Collection

Data were collected among participants from the four countries through semi-structured interviews, guided by interview guides. The development of these interview guides was inspired by the study aim, previous research conducted in The Netherlands [13], and the Technology Acceptance Model (TAM). Interview guides were tailored with specific questions to the macro, meso, and micro levels, as summarized in Appendix B. Data collection took place from November 2024 until February 2025.

Interviews were conducted either in person or online, depending on participants’ preferences. When meeting in person with participants, the researchers conducted interviews at the participants’ workplace or another location of the participant’s choice. Only the researcher(s) and the participants were present during the interview. In The Netherlands, 15 interviews were conducted with a total of 20 participants from macro and meso levels by P.T., S.V., or a Master’s student. Of these 15 interviews, 4 were conducted with 2 participants from the same organization, and 1 interview was conducted with 3 participants from the same organization. Some participants were familiar with S.V. or P.T. due to their work on the HealthEmove pilot, an EPHR for mobile populations in Amsterdam. In Denmark, 20 interviews were conducted with participants from all levels by M.B.N. or F.G. In Ghana, 40 interviews were conducted with participants from all levels by P.R.A. and two research assistants, and in Kenya, 10 consultations with stakeholders from macro and meso levels were conducted by S.K.P. and F.M.W. In all countries, there was no working relationship between the participants and the researchers. The interview guide was initially developed in English and translated by the researchers into Danish and Dutch. The interviews were primarily conducted in the official languages spoken in the respective countries: English in the case of Ghana and Kenya, Dutch in The Netherlands, and Danish in Denmark. The length of the interviews varied between 30 and 60 min.

### 2.5. Data Management and Analysis

All interviews were audio-recorded after informed consent was obtained from the participants. Transcription was performed with software assistance in The Netherlands (Sonix.ai) and Denmark (Good Tape). In Ghana and Kenya, professional transcribers were engaged to simultaneously transcribe the audio recordings. Transcripts were reviewed for accuracy and pseudo-anonymization.

While the transcripts of Denmark and The Netherlands were transcribed in Danish and in Dutch, respectively, data analysis was conducted in English for all transcripts to ensure consistency across countries.

Data analysis was informed by Braun and Clarkes’ iterative approach to thematic analysis [30] with their six phases of analysis adapted to fit the short research timeframe of this study (8 months).

The coding of the transcripts was conducted per country. Coding employed an iterative and collaborative approach. To increase researchers’ familiarization with the data, the transcripts were read several times by researchers coding the interviews, while taking contextual notes. Transcripts were coded manually and with the support of NVivo in Denmark, Ghana, and Kenya, and Atlas.ti in The Netherlands. A hybrid approach of inductive and deductive coding was applied to thematic analysis [31] in all countries. The Netherlands developed an initial code book, using a deductive approach informed by the TAM model and previous interviews conducted in The Netherlands [13]. Additional codes were subsequently added through inductive coding, based on the preliminary analysis of multiple interviews. This codebook was used as inspiration in each country. After an initial round of data coding by all countries, each country developed an adapted version of the codebook to fit its specific context. Thereafter, researchers discussed the output to better understand the findings from the different countries and to refine the codebook in each country.

Regular meetings were held to discuss the coding process and themes, and findings were shaped iteratively. Given the wide variety of codes and themes, a selection of the key themes within and across each country to investigate the perceived usefulness of EPHRs in improving access to and continuity of care for mobile populations across these four different health contexts was performed collectively through two interactive online meetings. Representatives of each country (P.T., M.B.N., F.G., S.K.P., P.R.A., and S.V.), facilitated by U.B.K., presented their key themes and reflected upon these themes with respect to their countries, collaboratively choosing the overarching themes and subthemes. This collaborative approach ensured that the final selection of themes was grounded in the realities of the different countries.

The deductive data coding for the micro level was informed by two core constructs from the TAM [20]: *perceived ease of use*—the degree to which a person believes that using a particular system would be free of effort—and *perceived usefulness*—the degree to which a person believes that using a particular system would enhance performance. We also considered *external variables*, referring to factors that influence perceived usefulness and perceived ease of use, but are not part of the core model itself [20]. Additionally, concepts of *social influence* and *facilitating conditions* from UTAUT [21] were used as a lens to incorporate meso perspectives. Social influence is the degree to which an individual perceives that important others believe he or she should use the new system, and facilitating conditions refer to the degree to which an individual believes that an organizational and technical infrastructure exists to support use of the system [21]. Recognizing that the TAM and UTAUT models focus on individual perspectives, the macro level perspectives were inductively coded to incorporate a multi-level context into the concept of perceived usefulness.

### 2.6. Ethical Considerations

All participants provided informed consent for their participation in each country. Ethical approvals in Ghana were granted by the Committee for Human Research and Publication Ethics (CHRPE/020/24), and in The Netherlands, by the Amsterdam University Medical Centre (number 2024.1052). In Denmark, ethical approval was not needed due to the observational nature of the study and national regulation in the country, as written in ACT 2018-05-13 no. 498 [32]. In Kenya, no ethical approval was deemed necessary for consultations with industry stakeholders from macro and meso levels, as no personal stories were being collected.

## 3. Results

### 3.1. Description of the Included Participants

In this study, 90 participants were included, 23 from the macro level, 39 from the meso level, and 28 from the micro level, of whom 21 were from mobile populations (Table 1). In total, four asylum seekers were interviewed in Denmark, and one UDM. In Ghana, 16 UDMs were interviewed.

While our study design emphasized a clear distinction between macro, meso, and micro perspectives, the preliminary data analysis revealed notable overlap between these levels. A possible reason for this overlap of insights was the dual role that some participants held, except for the mobile populations themselves. Some participants from macro and meso levels also worked as, for example, a HCP for mobile populations. Consequently, data was not analyzed within specific levels and themes presented that cut across the four levels. The full table of all included participants can be found in Appendix B.

### 3.2. Key Cross-Country Themes Derived from the Data

Key themes derived from the data related to the perceived usefulness of EPHRs in improving access to and continuity of care for mobile populations across the four countries included needs related to an increase in data access and ownership for mobile populations, opportunities for the improvement of access to and continuity of care and efficiency of care by means of an EPHR for mobile populations, and the challenges related to the lack of digital (health) literacy and concerns on data security of an EPHR. An overview of key themes and respective subthemes across the four countries is presented in Table 2.

#### 3.2.1. A Need for Increased Data Access and Ownership

A majority of study participants from all four countries stated that only a small portion of mobile populations, particularly among undocumented migrants, refugees, and asylum seekers, currently have ownership or access to their medical records. However, even people who have ownership of and access to their medical records often only have them on paper. Participants across all countries reported that mobile populations obtaining access to their medical records remains challenging, often due to a lack of legal frameworks for data sharing, limited willingness of HCPs to share the data, or time constraints faced by HCPs in facilitating medical record sharing.

Participants from all levels in the four countries emphasized the necessity for enhanced medical data sharing with mobile populations. Several participants in The Netherlands argued that it is ‘logical’ that mobile patients should become owners of their health records, and some participants even argued that it should be a requirement for these mobile groups, as illustrated by a senior advisor and GP from The Netherlands:


*“Yes, in The Netherlands, we are very used to healthcare providers communicating with each other. But why wouldn’t you make the patient the owner of the whole story? In fact, it is their data. So I think it is actually very logical.”*

*—P8, Macro and micro level, The Netherlands.*


Among undocumented migrants and asylum seekers in Ghana and Denmark, the importance of access to medical records was acknowledged. Asylum seekers from Denmark, for example, explained that if they were to leave Denmark suddenly, their medical records would be the first things they would take with them.

In line with the ongoing digitization of healthcare in The Netherlands and Denmark, participants noted that sharing medical data via an EPHR would “fit in the trends of today’s digitized society”. Furthermore, it was argued that an EPHR can empower individuals by providing them with greater control over their health information and facilitating self-directed care. A macro level participant in The Netherlands described how this “empowering” could happen with an EPHR serving as a form of legitimization:


*“So I really hope this is also a kind of legitimization […] almost like a city ID kind of idea. “Like, I exist, […]”. So I hope that this is also something empowering, and that around it you can also set some conditions, make some agreements.”*

*—P1, macro level, The Netherlands.*


In contrast to the ongoing digitization in Denmark and The Netherlands, digital systems in healthcare are still in a development phase in Ghana and Kenya. While a growing number of health facilities, especially those in the private sector, have digital medical record systems, some facilities still depend on paper records. In Ghana, a small but growing number of patients are beginning to request access to their health records. However, a healthcare provider from Ghana noted that patients are often less concerned with the method of accessing this information than with gaining access itself:


*“Although patients are beginning to enquire about [access to] their health records, it is often associated with visa acquisition, education or migration purposes. They do not care about the means, all they require is the health record.’*

*—P60, meso level, Ghana.*


Across both Ghana and Kenya, most participants favored digital medical record sharing; however, they stressed the need for practical, ethical, and legal guidelines to govern the sharing of (digital) health records with patients.

In Denmark and The Netherlands, participants were already more used to medical data sharing with ‘general’ patients, and therefore reflected on necessary practical guidelines and frameworks for EPHRs for mobile populations. They stressed that any EPHR implementation should ensure patient-controlled access, allowing individuals to decide what data is shared and with whom. Moreover, it was argued that patients should be able to access their health records independently, ideally across devices, and maintain ownership over their data.

#### 3.2.2. Recognized Potential of EPHRs to Improve Access to and Continuity of Care

Across all four countries, participants, especially HCPs, consistently emphasized that EPHRs could play a critical role in improving access to and continuity of care for disadvantaged mobile populations, a group for whom fragmented healthcare access is a recurrent challenge. As illustrated by a medical doctor in Denmark:


*“I believe it could improve continuity of care, especially if there is access to background information and medical history for the patient sitting in front of you. Due to time pressure, aspects like regular medication are often rushed through.”*

*—P42, micro level, Denmark.*


Continuity of care for mobile populations often depends on individuals manually transporting paper-based medical records—a practice observed across all countries. This approach is unsystematic and carries a considerable risk that patients may lose essential health information during their journey. Participants from the four countries expressed that an EPHR could ensure comprehensive, up-to-date patient histories that are accessible at any point of care, regardless of movement.

In Ghana, participants emphasized that EPHRs have the potential to close critical gaps in the healthcare system, particularly in the management of chronic diseases, by ensuring complete and accessible patient histories regardless of which provider a patient sees. In addition, stakeholders noted that EPHRs could facilitate faster emergency responses by providing immediate access to critical medical information, particularly for vulnerable groups such as the elderly or individuals unable to communicate their medical histories. As described by a meso level participant:


*“It will facilitate emergency response, the speed at which we need to introduce an intervention to somebody who is sick and cannot talk. […] Assuming such a system exists, and you move to one facility where you are not able to talk about your past history, the physician or whoever is attending to you can tell from your wallet that you are receiving this treatment.”*

*—P60, meso level, Ghana.*


In Kenya, where healthcare digitalization remains limited, macro level participants highlighted the government’s role in advancing continuity of care for mobile populations through the use of EHRs. Although not referring to personal health records specifically, participants emphasized the importance of accurate, accessible, and well-documented patient data as a foundation for effective follow-up, particularly when patients move between providers or care settings. As a participant from the Ministry of Health explained:


*“So, the use of a health record will definitely be very important because the data is now captured, and it cannot be lost. The other thing, of course it gives you the accuracy of the data. And for the clinician and for the patient who is seeking care, it assists you to have a proper record in terms of follow-up.”*

*—P22, macro level, Kenya.*


Participants from all four countries highlighted that an EPHR could enable mobile individuals to maintain their medical histories independently, supporting continuity of care by ensuring previous clinical information is accessible. This would facilitate better clinical decision-making, reduce the need for repeated diagnostic testing, and enhance the overall quality of care provided to mobile groups such as undocumented individuals and asylum seekers:


*“It is a fantastic idea; there are so many positive aspects, and it could provide at least a basic overview of these patients who are in the hands of so many different healthcare providers.”*

*—P41, micro level, General Practitioner, Denmark.*


Across all four countries, several participants stated that for EPHRs to truly support access to and continuity of care, an EPHR should be interoperable with existing health information systems, making sure that there is seamless integration into the healthcare workflow of HCPs. In contrast, in The Netherlands, it was also noted that linking an EPHR with all systems will be very complex. A possible solution is to view an EPHR as a ‘digital vault’ that is independent of existing systems. This would enable individuals, such as undocumented migrants who are not connected to traditional healthcare systems, to document their health information. For instance, they could upload pictures of their medical test results into their records, as illustrated by a GP:


*“Maybe you should see it that way [separate system] and not try to replicate something. […] Because it’s also a different group. […] If you’re a general practitioner and you’ve seen a patient [mobile population] and done lab tests, just print the results and ask the patient to take a photo of them in their app. That way, they have it as well.”*

*—P8, macro and micro level, GP and senior advisor, The Netherlands.*


Moreover, participants across all countries acknowledged that an easy-to-use and user-centric EPHR is a prerequisite for improving access and continuity of care for mobile populations.

#### 3.2.3. More Efficiency and Time Saving with EPHRs

Across all four countries, participants emphasized that EPHRs could significantly improve efficiency in healthcare provision for mobile populations.

Participants agreed that current paper-based personal health records or fragmented systems often lead to delays and inefficiencies, whereas an EPHR would allow new HCPs to quickly retrieve accurate medical information via the patient, saving both time and resources. Participants further highlighted that EPHRs could prevent the loss of crucial patient information and support better clinical follow-up.

In Denmark and The Netherlands, participants across all levels stressed that EPHRs could streamline transitions of patients in the healthcare system and reduce the need for repeated diagnostic procedures. As expressed by a member of the leading healthcare team at an asylum center in Denmark:


*“You would avoid an enormous number of unnecessary examinations, which would benefit the patient.”*

*—P38, meso level, Denmark.*


Danish participants further noted that faster access to the health data of mobile populations would ease their workload, especially during transitions from asylum centers to hospitals.

Similarly, Dutch participants from both macro and meso levels emphasized that if patients themselves could provide new doctors with access to their medical information, this would save significant time. Moreover, it was argued that it would reduce the administrative burden for HCPs, who otherwise have to manually request and transfer medical files for mobile populations who do not have a personal identification number.

In Ghana, participants expressed that EPHRs would reduce patient waiting times by enabling instant retrieval of laboratory results and medication histories, thus speeding up point-of-care decision-making:


*“Health professionals can retrieve patient histories, lab results, and medications instantly without needing to search paper files! This will reduce patient waiting times and speed up decision-making at the point of care.”*

*—P64, meso level, Ghana.*


In Kenya, participants also echoed these opinions, noting that many mobile and underserved patients often seek care from different facilities without bringing along a consistent medical history. EPHRs were believed to help reduce duplication of laboratory tests and consultations.

#### 3.2.4. EPHRs Require Education on Digital (Health) Literacy and Skills

A key challenge identified in all countries was the limited digital skills and (health) literacy among some mobile populations. Although many have access to a smartphone, this does not always translate into the ability to use and navigate digital health tools.

Several participants highlighted that low general literacy can hinder the effective use of an EPHR. A senior program officer from UNHCR emphasized that linguistic and digital diversity within this target group poses significant challenges to ensuring understanding and proper use of an EPHR:


*“You have to make sure that people understand how that [EPHR] works and how it should be and where it should be and in which language and things like that. That is of course complicated with this target group. In addition, you also have a very large […] part of this target group that is illiterate or, or cannot deal well with digital environments.”*

*—P10, macro level, The Netherlands.*


Cultural differences were also mentioned, particularly in The Netherlands, Denmark, and Ghana, as cultural background shapes how individuals interpret health information. As argued by a participant in The Netherlands, these variations in understanding of health and illness can pose significant challenges and increase the difficulty of understanding the content of an EPHR, especially across borders.

Another challenge mentioned in The Netherlands and Denmark was the importance of having the so-called ‘headspace’ to make use of an EPHR, as illustrated by a GP in The Netherlands:


*“Reading when you’re under so much stress, so focused on survival… Do you really think people would quickly find the time or want to spend time on that?”*

*—P17, meso and micro level, The Netherlands.*


Participants across all four countries stressed the importance of education and training, not only in how to use an EPHR but also in fostering broader awareness around digital transition in healthcare. As one meso level participant from Ghana explained:


*“When we started using the electronic health record system, patients, students, migrants, young and old still feel like we come to your hospital and we don’t get a paper prescription, it is not acceptable […] So they still need to be […] educated […] sensitized that we are moving from the paper to the electronic.”*

*—P59, meso level, Ghana.*


In Kenya, participants emphasized the need to involve and train community health promoters to support mobile populations in using digital health services. These trusted local actors and lay health providers based at the community level were seen as key to bridging literacy gaps and sustaining access to care for people on the move.


*“…just building that local capacity, preferably through community health promoters to be able to sustain health service delivery for the population on the move.”*

*—P27, meso level, Kenya.*


It was recommended in all four countries to introduce EPHRs in safe and familiar environments, such as through trusted community leaders or in social drop-in centers, and to educate new users on how to use and interpret EPHRs. An easy-to-use EPHR with a user-centered design developed in co-creation with end-users was seen as critical across the countries to ensure the EPHR’s usefulness and acceptability.

#### 3.2.5. Concerns About Data Privacy and Security of an EPHR

Participants across all four countries emphasized that strong data privacy and security safeguards are essential for the development and implementation of EPHRs. While many participants viewed an EPHR as promising, they also raised concerns about security risks associated with storing sensitive medical data of mobile populations in one location. This is illustrated by a Dutch coordinator of an NGO:


*“We’re talking about very vulnerable groups of people who essentially have all their data stored in a single file. I foresee… I can’t quite put my finger on what exactly the risks are, but I feel something about it that makes me uneasy.”*

*—P20, meso level, The Netherlands.*


Participants across the four countries also warned that individuals with access to their own EPHR might unknowingly or involuntarily share medical records with external entities, who could misuse their data. It was noted that some mobile individuals may not fully understand the implications of sharing sensitive health information, particularly when signing consent forms:


*“I think that—dangerous might be a big word—but I do think it carries risks. Someone who doesn’t realize what it is. The medical record. And who shares it with everyone. Either out of despair or ignorance or… So that is difficult, yes.”*

*—P12, meso level, The Netherlands.*


In Denmark, this is also addressed by individuals from the mobile population themselves. For example, one participant noted that UDMs working in sex work may be at high risk of unintentionally sharing their health information with the person they work for, which may result in significant negative consequences, including potential harm to the individual’s safety and well-being.

In contrast to these concerns about whether mobile populations would mistakenly share their medical records with untrustworthy external entities, in Denmark and Ghana, mobile populations expressed feeling uncomfortable with storing personal health data on their phone, citing a general distrust of mobile devices, particularly for sensitive medical information. As illustrated by an undocumented migrant from Ghana:


*“This will be a challenge for me. What if someone picks up my phone and finds my record? I have some health issues they don’t know about. I could lose my job. They will replace me with a healthier person.”*

*—P81, undocumented migrant, micro level, Ghana.*


In Denmark, there was also apprehension that medical records could be accessed without consent, for instance, if a phone was lost or taken.

In Ghana and Kenya, the absence of clearly implemented national data protection frameworks for individuals managing their own health data was seen as a key barrier to the safe implementation of EPHRs. While some legislation does exist in both countries to regulate the protection of personal data, participants pointed to a lack of clear, operational mechanisms—particularly when individuals, rather than health systems, are the primary custodians of their health records. This perceived regulatory gap raised concerns about data ownership, security, and accountability, especially in the context of mobile populations crossing national borders.

As one stakeholder in Kenya explained, although there are data protection regulations at the system level, it remains unclear how these would apply when health data is held directly by individuals:


*“But if you do that, who will be the owner of the system? […] How will it communicate with the country’s health systems and how is the issue of confidential personal data because it would boil down to security of information because an individual cannot hold the population data especially on health. The countries can because then there is already data protection regulations which if somebody does not use your data well you can sue them”.*

*—P29, meso level, Kenya.*


Moreover, only when data privacy is ensured will mobile populations be able to trust an EPHR, as outlined by a participant from Ghana:


*“A major concern with this is ensuring data privacy. How would patient data be protected? If people can be assured that their personal records will not be shared, then that will give them a high level of trust. No one would be happy to see their personal records being shared with someone else.”*

*—P54, macro level, Ghana.*


Participants from Kenya and Ghana argued that national policies on data security regarding an EPHR must align with international standards, given the mobility of populations across borders. In Denmark and The Netherlands, emphasis was put on developing EPHRs in line with the GDPR and not storing medical information on mobile phones, but rather on secure cloud-based systems.

## 4. Discussion

This multi-country qualitative feasibility study explored stakeholder perspectives on the perceived usefulness of EPHRs in improving access to and continuity of care for mobile populations living in disadvantaged circumstances in Denmark, Ghana, Kenya, and The Netherlands. Across all four countries, participants identified EPHRs as promising tools to improve data sharing, improve access to and continuity of care, reduce inefficiencies, and empower mobile individuals by enabling greater control over their health information. However, several key challenges were identified. These included low levels of digital and health literacy among mobile populations, limited system interoperability, and concerns about data privacy and security. Stakeholders also emphasized the need to educate mobile populations on how to use EPHRs effectively. Our study also identified intercountry differences. In Ghana, particular attention was given to preparing users for the broader transition toward digitized health records. Moreover, in both Ghana and Kenya, participants stressed the importance of establishing legal, ethical, and practical frameworks prior to the implementation of EPHRs. This illustrates both a shared perception of EPHRs’ potential and important differences in legal infrastructure and system readiness between these high-income and lower-middle-income countries. These findings contribute to the growing body of evidence that supports the potential of EPHRs to improve healthcare access and system responsiveness. They also demonstrate the importance of context-aware and inclusive implementation strategies, such as co-creation, community-based introduction, and user education, as emphasized both in our findings and in prior literature [14,33,34].

Our study shows that across the micro, meso, and macro levels and the different countries, EPHRs for mobile populations were widely seen as a promising tool to improve medical data sharing, to improve efficiency of care for both patients and HCPs, and to support access to and continuity of care within and across borders and healthcare systems. This aligns with prior studies suggesting that EPHRs may help mobile populations bridge gaps between fragmented services by acting as a portable source of their health history [11,12,19]. However, clear differences were seen in the ‘readiness’ of the countries for the implementation of an EPHR. Given the high level of digitalization in healthcare, the ability of registered citizens to already access their health record digitally, and national privacy laws in place in The Netherlands and Denmark, an EPHR was considered highly promising to provide mobile individuals access to their health data. In Ghana and Kenya, where health systems are not yet fully digitized, the focus should be on developing legal, ethical, and practical guidelines to support the introduction of EPHRs and protect individual ownership of sensitive health data.

While previous studies have emphasized the potential of EPHRs to improve access, continuity, and efficiency of healthcare [11,12], our study also highlights a unique characteristic of EPHRs, the emphasis on increased patient ownership, as EPHRs can function as patient-held tools. This distinguishes EPHRs from provider-owned records and positions the individual as the manager of their own health data. Indeed, many participants viewed this increased autonomy and self-determination as a clear benefit, particularly for mobile individuals who are known to lack formal access to national health systems [35]. However, others flagged an increased individual responsibility as a potential challenge and as an extra burden of managing sensitive health information; this is also expressed in previous literature [13,19]. Still, to the best of the researchers’ knowledge, no research has been conducted on the perceived attitude of mobile populations themselves on the responsibility of managing their own health records. Therefore, it is important to explore this further and to co-create an EPHR in collaboration with both HCPs and diverse groups within these mobile populations.

The concern of increased responsibility connects directly to another key study finding: the digital and health literacy barriers faced by many mobile individuals. Participants across countries pointed out that the ability to use and benefit from an EPHR presumes a certain level of digital fluency, linguistic competence, and psychological readiness—conditions that may not be present among those living in vulnerable or precarious circumstances. As noted in prior research, the ability to act autonomously in digital health environments requires a baseline of cognitive and emotional stability, which cannot be assumed in the case of displaced or undocumented populations [13,36]. Our study, indeed, highlighted the importance of educating mobile populations in both digital and health literacy. In Ghana, stakeholders particularly emphasized the need for educating patients on the shift from paper records to digital means. Previous studies also point towards the relevance of training HCPs on the effective use of digital health tools [37]. Ensuring meaningful use of EPHRs, therefore, requires more than access; it demands digital (health) training, culturally sensitive design, and trusted intermediaries who can support navigation and use, which was also reflected in our study.

Moreover, in The Netherlands, Denmark, and Ghana, participants noted that there are differences between people from different cultural backgrounds in the interpretation of medical information in an EPHR. This is supported by the literature—the study of Peterson [38] found that cultural and spiritual beliefs significantly influence understanding of illness and health. This underscores the importance of a culturally sensitive approach to EPHR implementation and development, especially given the cultural diversity among mobile populations. In addition to these cultural differences, participants, particularly in The Netherlands, highlighted that language barriers and low (health) literacy may limit understanding and effective use of EPHRs, underlining the need for culturally and linguistically sensitive design to ensure equitable access across diverse mobile populations In addition to different perceptions of health and illness across cultures, the literature indicates that there are also differences in legal frameworks across borders, such as contraception and abortion [39]. Some migrants may be reluctant to share health information due to concerns about discrimination, for instance, related to HIV status or other stigmatized conditions. Therefore, to ensure culturally sensitive use of EPHRs, it is crucial that future research explores user experiences and the technological potential for individuals to control access to specific parts of their health records.

Several participants across all countries and levels underscored that an EPHR for mobile populations must be interoperable with existing national health systems. This integration challenge is not unique to mobile populations nor to EPHRs and has been observed in broader EHR implementations globally [19,40]. Due to significant differences in digital infrastructure, integration could potentially progress more rapidly in Denmark and The Netherlands for asylum seekers than it would in Ghana and Kenya. Nevertheless, challenges remain even in the former, particularly around medical record sharing for individuals without a personal identification number. As a result, some Dutch participants suggested an alternative approach—developing an EPHR as a personal ‘data vault’ not directly linked to national systems but allowing individuals to store and manage their own documents and test results. This aligns with the HealthEmove pilot project and other existing EPHRs that function as standalone digital repositories [12].

Concerns about data privacy and security of an EPHR were a central theme across all four countries, particularly regarding the vulnerability of mobile populations and the potential misuse of sensitive health information. This is also reflected in broader debates in the digital health literature on balancing data accessibility with confidentiality and trust [19,41]. The importance of safeguarding health data is further emphasized by Matlin et al. [42] and supported by UNHCR [43], highlighting emerging risks related to internet connectivity, including cybersecurity and privacy breaches that may impact refugee status determination. Both our study and UNHCR [43] findings suggest that some individuals lack awareness of the value of their health data, underscoring the need for increased digital literacy and user education on data protection. Moreover, it underscores the necessity of very secure EPHR systems incorporating privacy-by-design.

Finally, in Ghana and Kenya, the absence of clearly implemented national data protection frameworks for individuals managing their own health data was seen as a critical barrier, underscoring the need for regulatory structures aligned with international standards. In contrast, participants in The Netherlands and Denmark emphasized the importance of compliance with GDPR and advocated for cloud-based solutions to avoid data storage on personal devices.

### 4.1. Implications for Further Research

We drew on the Technology Acceptance Model (TAM) [20] and the Unified Theory of Acceptance and Use of Technology (UTAUT) [21] to inform our methodological design. The TAM model was used as an inspiration for the development of our interview guides and initial coding framework. TAM was used broadly, while UTAUT informed our coding framework, specifically at the meso level. Both models emphasize determinants such as perceived usefulness, ease of use, and facilitating conditions (UTAUT) or external variables (TAM). Our findings reflect several of these core constructs. Participants frequently emphasized the importance of ease of use, integration with existing systems, and the need for support to improve digital and health literacy as critical enabling conditions for adoption. Additionally, our study identified socio-political and legal factors—such as exclusion based on lack of official identification and the absence of national guidelines for data sharing with individuals—that extend beyond the original scope of TAM and UTAUT. These insights suggest that while both models are valuable for understanding individual-level acceptance and adoption dynamics, they require expansion to adequately address multilevel insights, including structural barriers and contextual variability encountered in real-world, cross-cultural settings, particularly among mobile and underserved populations.

To realize the full potential of EPHRs for mobile populations, this study argues for a context-sensitive approach to their design and implementation. Policymakers and implementers should prioritize user-centered design, provide targeted education and training, and introduce EPHRs through trusted community structures. Aligning EPHR systems with legal data governance frameworks and ensuring interoperability are also crucial. Critically, future efforts must include co-creation with end-users—mobile populations, healthcare providers, and policymakers alike—throughout the design, implementation, and evaluation phases. Further research should explore the real-world use of EPHRs in mobile groups, focusing on longitudinal impacts on health outcomes, system efficiency, and the role of participatory design in fostering inclusive and sustainable digital health solutions. Moreover, it is relevant to consider EPHRs through the lens of implementation frameworks such as the Health Equity Implementation Framework [44], The Outcomes for Implementation Research [45], and the Consolidated Framework for Implementation Research (CFIR) [46], which can inform the co-creation, implementation, and integration of EPHRs in real-world healthcare contexts.

### 4.2. Strengths and Limitations

This study has applied a multi-level approach, including perspectives from ninety participants operating at various levels—macro, meso, and micro. Consequently, the findings reported are based on the insights of a diverse group of real-world experts, ranging from policy makers, project coordinators at NGOs, healthcare providers, and mobile populations themselves. This wide range of experts contributes to the information power of data [47] and enables a nuanced understanding of the real-world feasibility of EPHRs. This is particularly valuable given that scientific research on EPHRs for mobile populations is still an emerging field. Another strength of this study is its multi-country and multi-continent design, which included both data and researchers from four countries representing different healthcare digitization stages. This mimics the cross-country purpose of EPHRs and, therefore, allows gathering data that has relevance to a broader migration context.

This study nevertheless has some limitations that are important to state. No micro-level participants were interviewed in The Netherlands and Kenya due to time constraints. However, we included micro level participants from Denmark and Ghana, representing two countries that differ not only in the development of their general and digital healthcare systems, but also in cultural contexts that may shape perceptions of health, technology, and data use. Moreover, even though researchers had recurrent discussions about their analysis process and evolving findings, the need to summarize the highly rich datasets in each country into a collective paper might have led to an overrepresentation of Dutch and Danish insights due to the fact that digital health, and EPHRs in particular, are a more mature research topic in these settings. Furthermore, the uneven distribution of participants across categories and countries made direct comparisons more challenging. Therefore, the findings should not be interpreted as direct cross-country comparisons, but rather as a collective description of the exploratory nature of the study.

Additionally, while a clear methodological distinction was made between micro, meso, and macro level participants, researchers in the four countries, in some instances, had a different interpretation of the three levels. For example, The Netherlands labeled the UNHCR as macro, whereas Kenya labeled it as meso. Finally, as researchers from Denmark and The Netherlands translated the interview guide from English into Danish and Dutch themselves, it is recognized that translation and interpretation may be shaped by researchers’ perspectives.

### 4.3. Reflexivity

In line with a reflexive approach to thematic analysis and qualitative research more generally, the research team is aware of how researchers themselves function as research instruments and shape their qualitative findings. Reflexivity has, therefore, been a continuous practice during this study. Building on the researchers’ distinct academic and personal backgrounds, the team—in particular, researchers carrying out data collection and analysis—held recurrent cross-country discussions to navigate the complexity of analyzing and synthesizing data from four distinct contexts. P.T. and S.V. have been working with undocumented migrants and in the field of EPHRs for several years and are currently part of a team developing and testing HealthEmove, an EPHR for mobile populations, in Amsterdam. P.T.’s experience in the field of EPHR has guided this study design and data analysis from an early stage. P.T.’s deep engagement with EPHRs likely contributed to a more practice-oriented focus during study design and analysis. At the same time, this prior involvement may have introduced certain assumptions about the feasibility and value of EPHRs, particularly in relation to their technical functionality and integration into health systems. For M.B.N., S.K.P., and F.G., digital health and EPHRs is a new field of research. This outsider position has allowed for an open and critical stance toward the promises and challenges of EPHRs, not influenced by existing field norms or technological opportunities. M.B.N.’s experience as a GP, and particularly her previous work with socially vulnerable groups in Denmark, may have shaped her perspective as a researcher on the care of mobile populations, both by offering specific insights and introducing potential preconceptions. S.K.P. is a student of Clinical Psychology, and Dr. P.R.A. is a lecturer in Public Health with broad experience in qualitative research, which helped inform the methodology for this study. F.G. has research experience on migrant health from different global contexts. Her perspectives contributed to framing the broader global context of the study and informed key methodological considerations. This international experience also shaped the study’s analytical orientation, helping to identify cross-cultural dimensions of EPHR feasibility, but may have introduced a tendency to focus more on structural rather than individual-level barriers.

## 5. Conclusions

This study demonstrated that acceptance of and support for Electronic Personal Health Records (EPHRs) is widespread across diverse global settings and stakeholder levels (macro, meso, and micro). It further highlights the high perceived usefulness of EPHRs in enhancing data sharing, improving access to and continuity of care for mobile populations living in disadvantaged circumstances, and increasing patient ownership of medical records. At the same time, the study identifies technical, structural, and social barriers, including concerns about data security and varying levels of digital (health) literacy among mobile populations. While stressing the importance of cross-border data sharing through an EPHR for mobile populations, our findings also reveal that a one-size-fits-all approach for Denmark, Kenya, Ghana, and The Netherlands is unlikely to succeed given the differences in context, digital readiness, and the lack of privacy guidelines on patient ownership of medical data in Ghana and Kenya. With higher digital readiness in Denmark and The Netherlands, and a greater need for legal and infrastructural development in Ghana and Kenya, there is a need for context-aware EPHR development and implementation—EPHRs that are perceived as easy to use and supported by robust policy frameworks. Furthermore, education of mobile populations on digital health, medical records, and EPHR usage, alongside clear data protection measures, is essential for the acceptance and adoption of EPHRs. This study used the TAM and UTAUT models to explore the perceived usefulness of an EPHR as an indicator for user acceptance and intention to adopt an EPHR. However, to better understand digital innovations in complex real-world settings, we recommend further extending theoretical frameworks to address multi-level acceptance of health technologies. Specifically, we propose adapting theoretical models to expand from individual levels and incorporate a multi-stakeholder perspective, including legal factors and sociopolitical context. As global mobility continues to rise, the next steps in developing EPHRs for mobile populations should focus on further introduction, evaluation, and contextual adaptation of these systems. This should be performed in co-creation with stakeholders at all levels, ensuring alignment with cultural preferences and languages, and promoting equitable use of digital health to improve continuity and quality of care.

## Figures and Tables

**Table 1 ijerph-22-01363-t001:** Description of participants’ characteristics.

Country	Macro	Meso	Micro		Total
			**HCPs**	**Mobile Populations**	
The Netherlands	8	11	1	-	**20**
Denmark	5	4	6	5	**20**
Kenya	4	6	-	-	**10**
Ghana	6	18	-	16	**40**
	**23**	**39**	**7**	**21**	**90**

**Table 2 ijerph-22-01363-t002:** **Key** themes and subthemes per country.

Key Themes	Subthemes	Countries
A need for increased data access and ownership	Limited access to medical records and/or paper-based medical records	NL, DK, GH, KE
	Lack of practical guidelines for medical record sharing with patients	NL, DK, GH, KE
	Medical record sharing with patient is time-consuming	NL, DK, GH, KE
	Patient ownership of medical record is logical	NL
	Medical record ownership as empowerment	NL, DK, GH
	Health data ownership as a right	NL, DK
	Data ownership as a form of legitimization	NL
	Legal guidelines for EPHR needed	GH, KE
	Patient autonomy in an EPHR is important	NL, DK
Recognized potential of EPHRs to improve continuity of care	Ability to reduce fragmentation and loss of information	NL, DK, GH, KE
	Chronic care improvement	NL, DK, GH, KE
	Improvement of emergency response	GH, KE
	Available information for clinical decision-making	NL, DK, GH, KE
	Integration with existing health systems is needed	NL, DK, GH, KE
	EPHRs as a digital vault for the patient	NL
More efficiency and time saving with EPHRs	EPHR could save time	NL, DK, GH, KE
	Facilitation of patient transitions across healthcare settings	NL, DK, GH
	Reduced administrative burden	NL, DK, GH, KE
	Avoid duplicative examinations	NL, DK, GH, KE
	Shorter waiting times and faster clinical decisions	GH
EPHRs require education on digital (health) skills	Low digital skills of mobile groups	NL, DK, GH, KE
	Low (health) literacy skills of mobile groups	NL, DK, GH
	Different cultural understandings of health	NL, DK, GH
	Lack of mental capacity (‘headspace’) to engage with EPHRs	NL, DK
	Role of trusted local actors in supporting adoption	KE
	Introduction in safe, familiar environments with co-design	NL, DK, GH, KE
Concerns about data privacy and security of an EPHR	Risk of unintended data sharing by users	NL, DK, GH, KE
	Not fully grasping the sensitivity of health records	NL, DK
	Reluctance to store health data on personal digital devices	DK, GH
	Lack of national data protection laws for individuals using an EPHR	GH, KE
	Need for internationally aligned data protection standards	GH, KE
	Secure cloud-based storage as a preferred design	DK

NL = The Netherlands, DK = Denmark, GH = Ghana, KE = Kenya.

## Data Availability

Due to privacy concerns at the macro, meso, and micro levels, including from individuals representing mobile populations, interview transcripts are not available upon request. Several participants live in vulnerable circumstances, and sharing transcripts could risk exposing their legal status. Furthermore, participants did not provide informed consent for their transcripts to be stored in an online repository.

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
