# Peer review of "Exploring Stakeholders’ Perceptions of Electronic Personal Health Records for Mobile Populations Living in Disadvantaged Circumstances: A Multi-Country Feasibility Study in Denmark, Ghana, Kenya, and The Netherlands"

_ijerph, 2025, doi:10.3390/ijerph22091363_

Round 1
Reviewer 1 Report
Comments and Suggestions for Authors
I would like to thank the authors for the study that includes multi-country data and highlights the promise of EPHRs for improving continuity of care. The attention to context sensitivity and user education, coupled with the recognition that a one-size-fits-all approach is unlikely to succeed due to differences in digital readiness, legal frameworks, and infrastructure across these four countries, are all accurate observations.
I would like to offer some reflections regarding points of improvement for your consideration.
- The way the term “mobile populations” is applied in this particular study seems problematic. The study adopts the broad IOM definition but narrows the analysis to disadvantaged subgroups (displaced, refugees, asylum seekers, undocumented migrants), which are just one, vulnerable, segment of mobile populations. I see this creating a definitional inconsistency. The title and the methodology imply universality but in reality the study only examines a subgroup. Given that the study focuses on disadvantaged mobile populations, should it not say so explicitly in the title and throughout the text “Disadvantaged Mobile Populations” or “Vulnerable Mobile Populations”?
- The sample sizes, and particularly their uneven distribution across countries and categories, limit the comparability of findings.
- In the background section, the draft attributes the challenges associated with incomplete / inaccessible health records primarily to “mobile populations.” Should it also more explicitly distinguish between the effects of mobility and the disadvantages? Could it be that while displaced persons, undocumented migrants, and asylum seekers face barriers to continuity of care, these stem from structural, legal, and socioeconomic exclusion rather than from mobility? Clarifying this distinction would benefit the paper and remove the hit at conflating mobility with vulnerability. In its current form, I find that the argument risks overstating the causal link between being mobile and being disadvantaged.
- The conclusions primarily recapitulate what was already stated in the abstract and discussion, and is known. How do the authors see these advancing the debate? Would you consider adding points contributing to the development of a fresh theoretical or policy framework or typologies of EPHR implementation strategies tailored to different contexts?
Overall, this is a relevant line of inquiry to inform both research and practice, provided its conceptual clarity is enhanced.
Author Response
Dear reviewer, we would like to thank you for your compliments, comments and suggestions to improve the manuscript. Please find a response per comment below.
Comment 1: I would like to thank the authors for the study that includes multi-country data and highlights the promise of EPHRs for improving continuity of care. The attention to context sensitivity and user education, coupled with the recognition that a one-size-fits-all approach is unlikely to succeed due to differences in digital readiness, legal frameworks, and infrastructure across these four countries, are all accurate observations.
I would like to offer some reflections regarding points of improvement for your consideration.
The way the term “mobile populations” is applied in this particular study seems problematic. The study adopts the broad IOM definition but narrows the analysis to disadvantaged subgroups (displaced, refugees, asylum seekers, undocumented migrants), which are just one, vulnerable, segment of mobile populations. I see this creating a definitional inconsistency. The title and the methodology imply universality but in reality the study only examines a subgroup. Given that the study focuses on disadvantaged mobile populations, should it not say so explicitly in the title and throughout the text “Disadvantaged Mobile Populations” or “Vulnerable Mobile Populations”?
Response 1: We agree that our study focuses on disadvantaged mobile populations, which should be more clearly described in the manuscript. Within our research group, there has been a discussion about using the terminology ‘disadvantaged or vulnerable mobile populations’ given that certain circumstances might make mobile populations more disadvantaged or vulnerable. Therefore, we reached consensus to make use of the terminology “mobile populations living in disadvantaged circumstances”. This terminology is displayed in the title of the manuscript, in the abstract (page 1, line 31), discussion (page 16, line 643) and conclusion (page 20, line 851,) of the manuscript. Given the length of our chosen terminology, we decided to mention on page 2, lines 61-64: “This article focuses specifically on mobile populations that face more significant disadvantages, including displaced individuals, refugees, asylum seekers and undocumented migrants (UDMs), henceforth referred to as ‘mobile populations’”.
Comment 2: The sample sizes, and particularly their uneven distribution across countries and categories, limit the comparability of findings.
Response 2: Thank you for this comment. We agree that there is uneven distribution of participants across countries and categories, making comparison between the countries more difficult. Therefore, we decided not to conduct a comparison study, but rather describe findings across countries without direct comparison. We added in the manuscript a limitation related to the uneven distribution across the countries, and the lack of comparability, in page 19, paragraph “Strengths and limitations”, lines 802-805: “Furthermore, the uneven distribution of participants across categories and countries made direct comparisons more challenging. Therefore, the findings should not be interpreted as direct cross-country comparisons, but rather as a collective description of the exploratory nature of the study.”
Comment 3: In the background section, the draft attributes the challenges associated with incomplete / inaccessible health records primarily to “mobile populations.” Should it also more explicitly distinguish between the effects of mobility and the disadvantages? Could it be that while displaced persons, undocumented migrants, and asylum seekers face barriers to continuity of care, these stem from structural, legal, and socioeconomic exclusion rather than from mobility? Clarifying this distinction would benefit the paper and remove the hit at conflating mobility with vulnerability. In its current form, I find that the argument risks overstating the causal link between being mobile and being disadvantaged.
Response 3: Thank you for this valuable comment. We fully agree with your observation regarding the need to more clearly distinguish between the effects of mobility and structural disadvantage. In response, we have revised the introduction section to better clarify that the challenges mobile populations face in terms of continuity of care are not solely a result of mobility itself, but are compounded by intersecting structural, legal, administrative, and socioeconomic barriers. Specifically, we have added and rephrased lines 77-86, page 2 as follows: “While limited access to health records affects the general population, it presents a significant barrier to continuity of care for mobile populations. This is not only because mobile individuals frequently encounter new healthcare providers, but also due to the compounding effect of other disadvantages they face – including legal, administrative, economic, and systemic barriers.”
We hope this revision helps to clarify that mobility alone does not equate to vulnerability, and we appreciate your suggestion to refine this important distinction.
Comment 4: The conclusions primarily recapitulate what was already stated in the abstract and discussion and is known. How do the authors see these advancing the debate? Would you consider adding points contributing to the development of a fresh theoretical or policy framework or typologies of EPHR implementation strategies tailored to different contexts?
Response 4: Thank you for this interesting comment, we agree that the manuscript benefits from adding useful insights to our conclusion that advances the debate on EPHRs for mobile populations. Our study reports new insights on how acceptance and support for an EPHR for mobile populations living in disadvantaged circumstances is widely present in different settings of the globe (Denmark, Ghana, Kenya and the Netherlands), and among various stakeholder levels (macro, meso, micro). Therefore the following sentence is added: “This study demonstrated that acceptance of and support for Electronic Personal Health Records (EPHRs) is widespread across diverse global settings and stakeholder levels (macro, meso, and micro).”, page 20, lines 844-851. Moreover, we have proposed next steps that should be undertaken in developing an EPHR for mobile populations: “As global mobility continues to rise, the next steps in developing EPHRs for mobile populations should focus on further introduction, evaluation, and contextual adaptation of these systems. This should be done in co-creation with stakeholders at all levels, ensuring alignment with cultural preferences and languages, and promoting equitable use of digital health to improve continuity and quality of care.” page 20, lines 872-877. Additionally, we have proposed in the conclusion our insights to adapt theoretical models to include a multi-level stakeholder perspective: “This study used the TAM and UTAUT models to explore the perceived usefulness of an EPHR as an indicator for user acceptance and intention to adopt an EPHR. However, to better understand digital innovations in complex real-world settings, we recommend further extending theoretical frameworks to address multi-level acceptance of health technologies. Specifically, we propose adapting theoretical models to expand from individual levels and incorporate a multi-stakeholder perspective, including legal factors and sociopolitical context”, page 20, lines 864-870
Additional clarifications: Thank you for pointing out that the methods can be more clearly described. To improve the methodology section, the use of the TAM and UTAUT models have been more clearly described in paragraph 2.5, line 291-305. Moreover, thank you for pointing out that the figures and tables can be improved. We agree and have adjusted the tables to visually be more appealing in paragraph 3.1, table 1, page 7, line 321 – 322; paragraph 3.2, table 2, page 9, line 339-340 and in appendix III, page 23, line 1035-1036. Finally, to improve the readability of the results, several quotes in the results have been trimmed in line 373, 408-409, 438-441, 525, 617.
Reviewer 2 Report
Comments and Suggestions for Authors
Thank you for this well-conducted study. The study is well-written and addresses important global concerns, such as health equity for migrants and the challenges of sharing medical records across borders. However, the introduction feels a bit lengthy and could be improved by narrowing down the problem statement. Additionally, the early use of acronyms like UDMs and EPHRs may be unclear for readers who aren’t familiar with the terminology.
The methodology is strong and clearly laid out. Conducting 90 semi-structured interviews across four different countries adds both depth and valuable diversity to the study.
The findings are comprehensive and thoughtfully presented under key themes such as access to data, care continuity, system efficiency, digital literacy, and data privacy. That said, the repetition of some participant quotes could be trimmed to improve clarity.
The discussion makes good use of the data and connects well with existing literature and policy issues. It effectively highlights the promise of EPHRs, variations in digital infrastructure, and broader structural challenges. Still, the discussion could be strengthened by paying more attention to how cultural and language differences might affect the design and use of EPHRs in different settings.
Author Response
Thank you very much for taking the time to review this manuscript. Also, thank you for your kind words, compliments and comments to improve the quality of our manuscript.
Comment 1: Thank you for this well-conducted study. The study is well-written and addresses important global concerns, such as health equity for migrants and the challenges of sharing medical records across borders.
However, the introduction feels a bit lengthy and could be improved by narrowing down the problem statement.
Response 1: Thank you for this comment, we agree that the introduction feels a bit lengthy. Therefore, we decided to remove lines 127-151, on page 3-4 in section “Introduction” to the “Materials and Methods” section, lines 173-194, on page 4-5, paragraph “Study settings”. Moreover, we have deleted a sentence on lines 149-151, page 4, section “Introduction”. Important to note is that another reviewer stressed the importance of explaining the concepts of TAM and UTAUT in the introduction, therefore, the introduction contains an additional paragraph.
Comment 2: Additionally, the early use of acronyms like UDMs and EPHRs may be unclear for readers who aren’t familiar with the terminology.
Response 2: Dear reviewer, thank you for this useful comment, we agree with this comment, and we have removed multiple of the early acronyms in the introduction to help readers who are new to this topic getting familiar with the terms we use. See page 3, lines 102, 109, 115, 134 (EPHR) and page 2, line 115 (HCP) and page 3, lines 117 (UDM).
Comment 3: The methodology is strong and clearly laid out. Conducting 90 semi-structured interviews across four different countries adds both depth and valuable diversity to the study.
The findings are comprehensive and thoughtfully presented under key themes such as access to data, care continuity, system efficiency, digital literacy, and data privacy. That said, the repetition of some participant quotes could be trimmed to improve clarity.
Response 3: Thank you for this wonderful compliment. We agree that some of the quotes could be trimmed. Therefore, we trimmed quotes in line 373 (page 10), 408-409 (page 11), 438-441 (page 12), 525 (page 13), 617 (page 15).
Comment 4: The discussion makes good use of the data and connects well with existing literature and policy issues. It effectively highlights the promise of EPHRs, variations in digital infrastructure, and broader structural challenges. Still, the discussion could be strengthened by paying more attention to how cultural and language differences might affect the design and use of EPHRs in different settings.
Response 4: Thank you for this constructive comment. We agree that cultural and language differences can influence the design and use of EPHRs in different settings. To address this, we have revised the Discussion (lines 708-712, page 17) by adding the following text: "In addition to these cultural differences, participants – particularly in the Netherlands – highlighted that language barriers and low (health) literacy may limit understanding and effective use of EPHRs, underlining the need for culturally and linguistically sensitive design to ensure equitable access across diverse mobile populations." We have also strengthened the Conclusions based on this useful comment (lines 872–877, page 20) by adding: "As global mobility continues to rise, the next steps in developing EPHRs for mobile populations in disadvantaged circumstances should focus on further introduction, evaluation, and contextual adaptation of these systems. This should be done through co-creation with stakeholders at all levels, ensuring alignment with cultural preferences and languages, and promoting equitable use of digital health to improve continuity and quality of care."
Reviewer 3 Report
Comments and Suggestions for Authors
Overall this is an important topic on the global scale, and I applaud the authors for comparing two very different parts of the world for this research. I assume there is also a quantitative component to be presented elsewhere.
With that said, my concern is the theoretical constructs which are mentioned but not given much attention. Both the TAM and the UTAUT need to be described/defined fully in the introduction section. Even if the models were addressed in prior research (lines 181-182), this is a new manuscript. The data management and analysis section includes portions of the models, specifically perceived usefulness (TAM) and the social constructs (UTAUT). I am not familiar with research that picks parts of theoretical models to apply to a study, rather than the entire model. I believe the deviations should be explained somewhere or the theoretical constructs should not be included. In other words, using part of the construct to meet your needs without including all of it is not accurately representing the model. For the TAM, ease of use is never addressed.
Lines 434-438 addresses cultural differences, which I assume relate to the social influence portion of UTAUT. Again, in lines 585-597 cultural diversity is discussed, which I think is interesting in relation to adoption of personal health record systems, but the relation to UTAUT should be included (if my assumption that it is related is correct).
Lines 631-640 go into more detail about the TAM and UTAUT. Again, I feel that it is important to focus on the theoretical frameworks earlier in paper for these conclusions to make sense to the reader.
Author Response
Dear Reviewer, thank you very much for taking the time to review this manuscript. Additionally,
thank you for your kind words, compliments and suggestions to improve the manuscript. Please find our responses to the comments below:
Comment 1: Overall this is an important topic on the global scale, and I applaud the authors for comparing two very different parts of the world for this research. I assume there is also a quantitative component to be presented elsewhere.
With that said, my concern is the theoretical constructs which are mentioned but not given much attention. Both the TAM and the UTAUT need to be described/defined fully in the introduction section. Even if the models were addressed in prior research (lines 181-182), this is a new manuscript. The data management and analysis section includes portions of the models, specifically perceived usefulness (TAM) and the social constructs (UTAUT). I am not familiar with research that picks parts of theoretical models to apply to a study, rather than the entire model. I believe the deviations should be explained somewhere or the theoretical constructs should not be included. In other words, using part of the construct to meet your needs without including all of it is not accurately representing the model. For the TAM, ease of use is never addressed. Lines 434-438 addresses cultural differences, which I assume relate to the social influence portion of UTAUT. Again, in lines 585-597 cultural diversity is discussed, which I think is interesting in relation to adoption of personal health record systems, but the relation to UTAUT should be included (if my assumption that it is related is correct). Lines 631-640 go into more detail about the TAM and UTAUT. Again, I feel that it is important to focus on the theoretical frameworks earlier in paper for these conclusions to make sense to the reader.
Response 1: Dear reviewer, thank you for the constructive feedback on our work. We understand your concerns; however, our study focussed on the ‘perceived usefulness’, not on the ‘actual usage of an EPHR’. Therefore, from the TAM model ‘external variables’, ‘perceived ease-of-use’ and ‘perceived usefulness’ are used, which are in line with the ‘performance expectancy’ and ‘effort expectancy’ of the UTAUT model. The TAM model was used as an inspiration for the interview guide, however, due to multiple feedback rounds several questions have been removed or rephrased. The TAM model was used as an inspiration for the initial codebook, and the UTAUT was used to incorporate ‘meso’ perspectives specifically focussed on ‘social influence’ and ‘facilitating conditions’ when the TAM model did not seem sufficient to incorporate meso-level perspectives. However, we also learned during our study, that TAM and UTAUT, which focus on individual-level perspectives, are not sufficient frameworks to understand the perceived usefulness of EPHRs for mobile populations. A paragraph in the introduction is added to introduce the models, page 4 lines 151 – 159: ”This study draws on two widely recognized theoretical frameworks on technology acceptance: the Technology Acceptance Model (TAM) [17] and the Unified Theory of Acceptance and Use of Technology (UTAUT) [28]. While both models focus on individual-level acceptance of technology, they emphasize key constructs such as perceived usefulness (TAM) or performance expectancy (UTAUT), and perceived ease of use (TAM) or effort expectancy (UTAUT), as well as social influence and facilitating conditions (UTAUT). Our study integrates these core concepts to explore multi-level opportunities and challenges related to the perceived usefulness of Electronic Personal Health Records (EPHRs) among mobile populations across diverse settings.”
We indeed see that we did not fully describe the concept of ease-of-use (TAM) nor effort expectancy (UTAUT) clearly in the manuscript text, however, these concepts are used in the analysis of our data and in the result section to explore how they relate to the perceived usefulness of an EPHR, which was the main focus of our study. Therefore, we have added additional information in the method section, lines 291-305, page 7 paragraph 2.5, “The deductive data coding for micro level was informed by two core constructs from the TAM [20]: Perceived ease of use - the degree to which a person believes that using a particular system would be free of effort- and perceived usefulness - the degree to which a person believes that using a particular system would enhance performance. We also considered external variables, referring to factors that influence perceived usefulness and perceived ease of use, but are not part of the core model itself [20]. Additionally, concepts of social influence and facilitating conditions from UTAUT [21] were used as a lens to incorporate meso perspectives. Social influence is the degree to which an individual perceives that important others believe he or she should use the new system, and facilitating conditions refers to the degree to which an individual believes that an organizational and technical infrastructure exists to support use of the system [21]. Recognizing that the TAM and UTAUT models focus on individual perspectives, the macro level perspectives were inductively coded to incorporate a broader level context into the concept of perceived usefulness”, and in the result section lines 471-473 (page 12, paragraph 3.1.2) on the ease of use, as a precondition for improved access and continuity of care when implementing an EPHR: “Moreover, participants across all countries acknowledged that an easy-to-use, and user-centric EPHR is a prerequisite for improving access and continuity of care for mobile populations”. Additionally, few words are added on the ease-of-use on line 562 (page 14, paragraph 3.1.4): “An easy-to-use EPHR with an user-centered design and developed in …”.
Furthermore, in the conclusion we have reflected on the lack of a multi-level stakeholder perspective in the TAM and UTAUT and have proposed an extension in lines 864-870, page 20: “This study used the TAM and UTAUT models to explore the perceived usefulness of an EPHR as an indicator for user acceptance and intention to adopt an EPHR. However, to better understand digital innovations in complex real-world settings, we recommend further extending theoretical frameworks to address multi-level acceptance of health technologies. Specifically, we propose adapting theoretical models to incorporate a multi-stakeholder perspective, including legal factors and sociopolitical context”.
Round 2
Reviewer 3 Report
Comments and Suggestions for Authors
Thank you for addressing my concerns about the use of the theoretical constructs of the TAM and UTAUT. I believe the paper is much improved with the additional information.